# Enhancing Missense Variant Pathogenicity Prediction with MissenseNet: Integrating Structural Insights and ShuffleNet-Based Deep Learning Techniques

**DOI:** 10.3390/biom14091105

**Published:** 2024-09-02

**Authors:** Jing Liu, Yingying Chen, Kai Huang, Xiao Guan

**Affiliations:** 1College of Information Engineering, Shanghai Maritime University, Shanghai 201306, China; jingliu@shmtu.edu.cn (J.L.); chenyingying@stu.shmtu.edu.cn (Y.C.); 2School of Health Science and Engineering, University of Shanghai for Science and Technology, Shanghai 200093, China; hk@usst.edu.cn; 3National Grain Industry (Urban Grain and Oil Security) Technology Innovation Center, Shanghai 200093, China

**Keywords:** missense variant, pathogenicity prediction, deep learning, attention mechanisms

## Abstract

The classification of missense variant pathogenicity continues to pose significant challenges in human genetics, necessitating precise predictions of functional impacts for effective disease diagnosis and personalized treatment strategies. Traditional methods, often compromised by suboptimal feature selection and limited generalizability, are outpaced by the enhanced classification model, MissenseNet (Missense Classification Network). This model, advancing beyond standard predictive features, incorporates structural insights from AlphaFold2 protein predictions, thus optimizing structural data utilization. MissenseNet, built on the ShuffleNet architecture, incorporates an encoder-decoder framework and a Squeeze-and-Excitation (SE) module designed to adaptively adjust channel weights and enhance feature fusion and interaction. The model’s efficacy in classifying pathogenicity has been validated through superior accuracy compared to conventional methods and by achieving the highest areas under the Receiver Operating Characteristic (ROC) and Precision-Recall (PR) curves (Area Under the Curve and Area Under the Precision-Recall Curve) in an independent test set, thus underscoring its superiority.

## 1. Introduction

In the human genome, approximately three million single nucleotide variations (SNVs) are present, representing one-thousandth of all genomic bases. SNVs within coding regions are classified either as synonymous or nonsynonymous mutations. It has been found that nonsynonymous SNVs (nsSNVs), which account for about 90% of all genetic variation types in humans, vary in their effects on protein function and stability by altering both genotypes and phenotypes [1]. Studies have revealed that nearly one-third of nsSNVs adversely affect human health, potentially leading to various diseases [2]. In the field of nutrition, poor nutritional status can exacerbate inflammatory responses and disrupt metabolic pathways, increasing the risk of cardiovascular diseases and diabetes [3,4]. In cancer research, certain genetic variations have been identified as increasing the risk of developing breast cancer and colorectal cancer [5,6]. Additionally, genetic variations are linked to the occurrence of rare diseases such as Huntington’s disease and cystic fibrosis [7,8].

Among the more than four million missense mutations identified, only approximately 2% have been definitively classified as either pathogenic or benign [9]. The classification of the remaining variants, which have ambiguous effects, remains a significant challenge in human genetics. As the number of recognized single nucleotide variations (SNVs) continues to grow, distinguishing effectively between pathogenic mutations and those that are functionally benign has become increasingly complex. Traditionally, laboratory experiments have been integral to assessing the impacts of SNVs on protein function and their roles in disease etiology. However, these methods are typically labor-intensive and time-consuming, and they do not consistently yield reliable predictions regarding the pathogenicity of mutations [10].

Computational methods and models are being actively developed by researchers to more effectively distinguish pathogenic mutations from functionally benign substitutions, aiming to predict the impacts of single nucleotide variations (SNVs) on protein function. A variety of computational tools, such as MutPred2 [11], nsSNPAnalyzer [12], PolyPhen-1 (PPH-1) [13], PolyPhen-2 (PPH-2) [14], SNAP [15], PANTHER [16], PhD-SNP [17], SIFT [18], and SNPs&GO [19], are employed to assess how amino acid substitutions may affect protein functionality. These methods incorporate factors, including protein structure, evolutionary conservation, and functional domains, which assist in identifying SNVs potentially associated with disease-related functional changes. Furthermore, advances in artificial intelligence and machine learning have led to the increased use of these tools in analyzing vast genomic datasets, thereby enhancing the speed of disease research and drug development by facilitating more rapid predictions of SNV pathogenicity.

The structural context of modified amino acids offers critical insights into their impacts on protein function [9]. However, most prevalent predictive tools have not yet fully integrated 3D structural data of proteins. This limitation is partly due to the lack of experimental structures for approximately 80% of protein residues [20,21]. PolyPhen-2 was a pioneer in incorporating such data, utilizing metrics such as residue accessibility, shifts in hydrophobic tendencies, and crystallographic B factors for proteins with available structures [14]. Protein structures provide invaluable data that are essential for elucidating biological processes and aiding in structure-based drug development or targeted mutagenesis. With the introduction of AlphaFold 2 [22], precise three-dimensional protein structures can now be determined without reliance on experimental data. Tools like Alphscore [23] and AlphaMissense [9] have opened new avenues for analyzing pathogenic missense mutations by leveraging structural features predicted by AlphaFold2. The adoption of AlphaFold-derived systems to integrate structural insights has greatly enhanced clinical annotations, the identification of novel disease mutations, and experimental benchmarks. However, these approaches also reveal that focusing exclusively on protein structure may overlook other vital biological details such as genetic background, epigenetic modifications, or the impacts of mutations in non-coding regions.

Currently, methods for predicting pathogenicity predominantly employ traditional machine learning techniques that rely heavily on the extraction of features such as the physicochemical properties of proteins, genetic information, and evolutionary conservation. Tools like PolyPhen-2 and SIFT, for example, evaluate the potential impacts of mutations by analyzing genetic sequence conservation and alterations in protein structure. A significant advantage of these tools is their ability to produce reasonable predictions without extensive labeled data. However, these methods generally depend on intricately designed features, which limit their ability to fully capture the inherent complexity of biological systems. Furthermore, machine learning models often require a clearly defined problem framework and specific input features, which may not always be feasible due to the variability inherent in biological diversity [24,25].

We propose new strategies to enhance the prediction of pathogenicity in missense mutations. Our approach includes developing innovative deep-learning models that leverage extensive datasets to boost predictive accuracy. These models are designed to robustly learn and elucidate complex relationships between protein structure and function. Additionally, we aim to integrate predictive scores from AlphaFold2, an advanced protein structure prediction tool, as a key feature. This integration is expected to refine the accuracy of our predictions by utilizing AlphaFold2′s ability to provide critical structural insights essential for assessing the pathogenicity of missense mutations. Figure 1 illustrates the overall structure of the paper.

## 2. Feature Extractions

### 2.1. Commonly Used Predictive Features

With the widespread adoption of next-generation sequencing technologies in genetic medicine, a diverse array of computational methodologies using machine learning or deep learning has been developed to predict the pathogenicity of missense variants, significantly advancing research efforts. These methodologies are broadly categorized into three types based on their predictive approaches: sequence-based prediction methods, methods integrating structural and functional features, and ensemble tools [26,27,28].

Sequence-based prediction methods assess the potential pathogenicity of missing variants by examining the sequence of nucleotides or amino acids in a gene or protein. These methods rely heavily on the analysis of DNA or protein sequences to detect variations that may alter function or stability.

Methods integrating structural and functional features are utilized to determine the pathogenicity of missense variants by analyzing the structural and functional attributes of proteins synthesized from transcription. This analysis includes examining aspects such as secondary structures.

Ensemble tools combine results from multiple pathogenicity prediction tools, establishing unique evaluation criteria and incorporating advanced techniques such as machine learning and deep learning. This ensemble approach significantly enhances both prediction accuracy and reliability.

Features commonly used to differentiate pathogenic from prevalent nsSNVs were sourced from dbNSFP v4.2a. Initially, features underwent preliminary screening based on the coverage percentages of pathogenic and common nsSNVs, as determined by InMeRF’s [29] analysis across 37 tools. Subsequently, Pearson correlation coefficients were calculated to assess the relationships between predictive factors and the clinical labels (pathogenicity) of the variants. Features deemed to be of very low significance were excluded.

The employed features are outlined in the table below, with sequence-based prediction methods classified as Category A, methods integrating structural and functional features as Category B, and ensembletools as Category C. Specific feature classifications are presented in Table 1.

### 2.2. AlphFold2 Structural Features

Protein structures offer extensive insights into the three-dimensional conformations of proteins, including inter-residual distances, interactions, and the organization of structural domains. Understanding this structural context is crucial for assessing the impact on protein function and evolutionary constraints. Variations that destabilize the protein structure or impair its function may be subject to evolutionary constraints. Notably, missense mutations located in the core of the tertiary structure are more frequently associated with diseases than those on the surface [45]. The AlphaFold2 team has successfully predicted protein structures by inputting protein sequences, achieving results that closely match experimentally determined structures, especially in terms of topology and local conformation. These predictions from AlphaFold2 have significantly resolved the ‘ambiguity’ issue in structural biology, enhancing both the reliability and accuracy of predicted protein structures. These models form a foundation for understanding various aspects of protein biology.

The highly precise structural models from AlphaFold2, along with the capability to learn evolutionary constraints from related sequences, are leveraged by AlphaMissense to predict the pathogenicity of missense variants. This innovative approach combines insights from protein structure and evolutionary data, providing a more accurate and comprehensive perspective for predicting the pathogenicity of missense variants.

In this study, protein structures were predicted using AlphaFold2, and the FEATURE framework [46] was integrated to compute a series of features related to protein sequences and structures. These features include interactions within different radii (e.g., 0 or 6 Å), such as aromatic-aromatic, aromatic-sulfur, backbone-side chain, side chain-side chain hydrogen bonds, and hydrophobic interactions. Graph-theoretic features such as degree of neighbors, betweenness centrality, eigenvector centrality, and weighted degree were calculated using a networkx to reflect the importance of residues within the protein structure. Additionally, the residues’ hemispheric exposure and solvent accessibility were assessed to evaluate their exposure to the protein surface. The FEATURE framework was also employed to calculate the charge and hydrophobicity of residues, and a weighted amino acid network provided weighted averages of neighboring residues. The predictive quality of residues and overall protein structures was assessed using AlphaFold2’s pLDDT scores. Finally, substituted amino acids were categorized into charged, hydrophobic, polar, acidic, and basic groups, and it was noted whether they belonged to these categories. All these features, through combined computation and graph-theoretical analysis, aid in better understanding the stability, functionality, and potential mutational impacts on protein structures [16].

## 3. Data Processing

### 3.1. Data Set Composition

Three datasets of missense variants have been compiled from the gnomAD [47] (version 3.1) and ClinVar [48] (version 20210131) databases to support the training, validation, and testing phases of this study. Clinically annotated non-synonymous variants are categorized as ‘Benign’ or ‘Pathogenic’ to represent benign and pathogenic variants, respectively. Eighty percent of these non-synonymous variants were randomly selected from proteins, with gnomAD variants designated for the training set and ClinVar variants not present in gnomAD allocated to the validation set. The remaining 20% of proteins containing ClinVar variants have been designated for the test set. To enhance the accuracy and reliability of the research data, all variants in the ClinVar dataset with a review status of less than one star have been excluded [23].

### 3.2. Data Acquisition

For the training, validation, and test sets, the total number of variants with missing predictors was documented, identifying those with 4, 5, 6, or more missing elements. To maintain the integrity of the overall dataset, variants with five or more missing predictors were excluded from further analysis. The remaining missing values were addressed through mean imputation.

Following site selection and missing value imputation, it was retained that the training dataset contained 301,353 mutation sites, whereas the validation and test sets included 24,978 and 9978 sites, respectively. Upon examination, the ratios of positive to negative samples in the validation and test sets were found to be between 1.5 and 2—a range deemed acceptable for experimental purposes. Conversely, the training set exhibited a ratio of approximately 3.84:1, indicating a significant imbalance that necessitates further corrective measures.

### 3.3. Data Imbalance Treatment

Dataset imbalance is a common challenge in multi-label classification tasks. To address imbalances in missense variant classifications, undersampling is frequently employed [49]. In this research, a data augmentation strategy was implemented to increase the number of samples in the minority class.

For any given minority class sample $x_{i}$, its neighboring region $R\left(x_{i}x_{k-q}\right)$ is defined as follows:(1)$$R\left(x_{i}x_{k-q}\right)=\left\{x:d\left(x_{i},x\right)\leq d\left(x_{i},x_{k-q}\right)\right\}$$

Here, $d\left(x_{i},x\right)$ represents the distance between the sample $x_{i}$ and another sample x, with $x_{k-q}$ being the k nearest neighbor of the minority class sample.

After identifying the candidate region, synthetic samples are generated using an interpolation method as follows:(2)$$x_{syn}=x_{i}+\gamma \left(x_{as}-x_{i}\right),\gamma \in \left[0,1\right]$$

In this formula, $x_{as}$ acts as an auxiliary seed sample, one of the neighbors of the minority class sample, and γ is a random number within the range [0, 1] [50].

The final training dataset comprised 363,594 mutation sites.

## 4. Construction of the Model

### 4.1. ShuffleNet

In contemporary convolutional neural network (CNN) architectures, advanced models such as Xception [51] and ResNet [52] achieve a balance between computational efficiency and expressive power through the integration of innovative convolutional technologies. Grouped convolution, a technique known for its efficiency, is utilized to reduce both the number of parameters and the computational complexity of the model by dividing the input feature channels into several groups. In this arrangement, computations are processed independently by each group using its own set of convolutional kernels. Although this method significantly reduces the computational burden, it is observed that inter-channel interactions are limited because the output of each group is only related to inputs from that same group, thereby constraining the network’s ability to learn holistically.

In the ResNet architecture, pointwise convolution predominates, accounting for more than 90% of the network’s operations, whereas grouped convolution is mainly employed within the smaller 3 × 3 convolution layers. This configuration is especially prevalent in compact networks operating under computational constraints, where the high costs associated with pointwise convolutions limit the number of channels that can be sustained, potentially impacting model performance.

To address this limitation, a novel approach is introduced by ShuffleNet [53], which incorporates a channel shuffling mechanism that optimizes convolutional operations. This method allows for the effective exchange of feature map information across different channel groupings by implementing a random and uniform shuffling of channels after grouped convolution. This strategy not only avoids additional computational costs but also significantly improves the model’s information flow and overall expressive capabilities. It ensures that the input features of each convolutional group are enriched with output features from various groups. Figure 2 depicts the core design of ShuffleNet V2, which is founded on four guiding principles crucial to its architecture and efficiency. Each principle is visually represented to detail how its contribution to the model’s performance and optimization is facilitated.

ShuffleNet V2, utilized as the primary backbone network in this research, is an advanced iteration of ShuffleNet V1, specifically adapted for scenarios with limited computational resources. This network is crafted based on four fundamental principles aimed at maximizing performance within constrained computational budgets. Key technical enhancements include the following:

Within each basic unit, the input feature channels are divided into two branches to promote parallel processing. One branch is designed to minimize network fragmentation and boost parallelism, while the other employs three convolutions with matching input and output channel dimensions to equalize the channel width and reduce the memory access cost (MAC) [54].

To reduce the computational overhead introduced by grouped convolutions, the grouping operation in the 1 × 1 convolution layers has been removed. In its place, two separate 1 × 1 convolutions are used to substitute the group-wise operations, thereby decreasing MAC.

It is assumed that the number of groups in a group convolution is g as follows:(3)$$B=\frac{hwc_{i}c_{o}}{g}$$
(4)$$\begin{array}{cc} MAC & =hw\left(c_{i}+c_{o}\right)+\frac{c_{i}c_{o}}{g}\\ & =hwc_{i}+\frac{Bg}{c_{i}}+\frac{B}{hw} \end{array}$$

It is observed that as the number of groups g increases, the MAC also increases, given that the number of floating-point operations B remains constant.

In this context, it is assumed that for a 1×1 convolution, the input feature size is $c_{i}\times h\times w$, $c_{o}$ represents the number of output channels, and B denotes the FLOPs.
(5)$$B=hwc_{i}c_{o}$$
(6)$$MAC=hw\left(c_{i}+c_{o}\right)+c_{i}c_{o}$$

When B remains constant, the arithmetic mean inequality suggests: (7)$$MAC\geq 2\sqrt{hwB}+\frac{B}{hw}$$
when $c_{i}=c_{o}$, then the inequality is satisfied and the MAC is minimized.

In the network’s branching process, the additive operation (Add) is replaced by a concatenation (Concat), followed by channel reordering. This modification ensures consistency in channel count and enhances computational efficiency.

Instead of using Channel Split, the downsampling unit performs direct downsampling from the original input. Each branch applies a depthwise convolution with a stride of 2 for downsampling, followed by a 1 × 1 convolution to adjust the channel counts, thus optimizing performance.

These strategic enhancements guarantee that ShuffleNet V2 delivers outstanding performance under computational constraints and offers significant advantages over competing models. The sophisticated network design of ShuffleNet V2 enables MissenseNet to substantially enhance the predictive accuracy of pathogenicity in missense variants, providing robust technical support for clinical decision-making.

### 4.2. Squeeze-and-Excitation Module

The Squeeze-and-Excitation (SE) module is strategically employed to enhance the performance of convolutional neural networks by explicitly modeling dependencies among channels, thereby significantly improving the network’s representational capabilities. The fundamental aspect of the SE module involves recalibrating each channel’s feature response through a learning process, effectively leveraging the network’s inherent feature information [55].

In this phase, each channel in the feature map undergoes spatial information compression via a global average pooling operation, producing a channel descriptor. For a feature map *U* of dimensions H×W×C, where H, W, and C represent the height, width, and number of channels, respectively, the output $Z_{c}$ is computed as follows:(8)$$z_{c}=\frac{1}{H\times W}\sum_{i=1}^{H}\sum_{j=1}^{W}u_{c}\left(i,j\right)$$
where $u_{c}\left(i,j\right)$ denotes the feature value at position (*i*, *j*) in channel *c*, while $z_{c}$ denotes the globally spatially compressed feature of channel c.

Following the squeeze step, this stage utilizes a fully connected layer to learn the dependencies between channels and recalibrate the channel responses. The compressed feature *z* is processed through dimensional reduction and expansion in a fully connected layer, with subsequent application of ReLU and sigmoid activations generating the channel weights *s*:(9)$$s=\sigma \left(g\left(z,W\right)\right)=\sigma \left(W_{2}\delta \left(W_{1}z\right)\right)$$

In this description, g functions as the fully connected layer’s operation, with $W_{1}$ and $W_{2}$ serving as its weight parameters. The ReLU activation function is denoted by δ, and the sigmoid function by σ, which restricts the weights $S_{c}$ of each channel to the range [0, 1]. Consequently, the output feature map $\tilde{U}$ is derived by multiplying the original input U with the learned weights s:(10)$${\tilde{U}}_{c}=s_{c}\cdot u_{c}$$

This methodology enables network efficiency to be enhanced by preferentially allocating resources to information-rich channels while attenuating those with scant data, thereby elevating overall network performance. Substantial improvements in performance have been consistently documented across a broad spectrum of standard visual tasks and network architectures following the integration of the SE module.

### 4.3. Propose the Model MissenseNet

To address the issue of low classification accuracy in predicting the pathogenicity of missense variants, this study introduces a novel missense variant classification model named MissenseNet. This model is built on an enhanced ShuffleNet V2 framework and incorporates an attention mechanism to improve diagnostic accuracy. The architecture of MissenseNet features a composite structure that includes an encoding network, a decoding network, and a classification network.

The encoding network is primarily composed of depth-wise separable convolutions coupled with the nonlinear activation function ReLU. It is specifically designed to compress advanced features through dimensionality reduction, effectively minimizing the redundancy of input data. Consisting of a single depthwise separable convolution unit, the decoding network is tasked with transferring the encoded features to the classification network [56]. The classification network, which integrates multiple ShuffleNet modules, is optimized to excel in extracting key features. This integrated approach significantly enhances the model’s ability to accurately classify pathogenic variants, thereby improving its diagnostic capabilities.

## 5. Model Evaluation Index

This article presents a suite of metrics specifically designed to intuitively evaluate the classification efficacy of the model. The metrics employed include accuracy, recall, precision, F1 score, Matthews correlation coefficient (MCC), and balanced accuracy. To accurately capture the model’s predictive performance, the mean of the accuracy averages, derived from ten instances of tenfold cross-validation and, hereafter, referred to as ACC, has been utilized to represent the model’s overall accuracy. In addition to these standard evaluation metrics, the Receiver Operating Characteristic (ROC) curve and the Precision-Recall (PR) curve, along with their corresponding areas under the curve (AUC and APUC, respectively), have also been used to thoroughly assess the model’s performance. These metrics provide detailed insights into the model’s capabilities across various operational thresholds, offering a comprehensive evaluation of its effectiveness.
(11)$$Accuracy=\frac{TP+TN}{TP+FP+TN+FN}$$
(12)$$Precision=\frac{TP}{TP+FP}$$
(13)$$Recall=\frac{TP}{TP+FN}$$
(14)$$F1=2\cdot \frac{Precision\cdot Recall}{Precision+Recall}$$
(15)$$MCC=\frac{\left(TP\times TN\right)-\left(FP\times FN\right)}{\sqrt{\left(TP+FP\right)\left(TP+FN\right)\left(TN+FP\right)\left(TN+FN\right)}}$$
(16)$$\begin{array}{cc} Balance & Accuracy \end{array}=\frac{1}{2}\left(\frac{TP}{TP+FN}+\frac{TN}{TN+FP}\right)$$

## 6. Results

### 6.1. Feature Ablation Experiments

A series of feature ablation studies were conducted to evaluate the impact of various feature categories on the performance of our model. These investigations focused on four feature sets: sequence-based prediction methods (Category A), hybrid structure and function-based prediction methods (Category B), integrated prediction tools (Category C), and structural features predicted by AlphaFold2 (Category D). Missing values within these categories were uniformly imputed using the mean substitution method. All performance metrics were derived from the average results of tenfold cross-validation.

As presented in Table 2, when evaluated as standalone predictors, sequence-based features (Category A), hybrid structure-function features (Category B), and structural features predicted by AlphaFold2 (Category D) were found to demonstrate limited efficacy. In stark contrast, it was shown that features in Category C markedly outperformed these groups in terms of accuracy, precision, and area under the ROC curve (AUC). Classification accuracy of 90.48% and an AUC of 0.9625 were achieved by features in Category C. This enhanced performance is attributable to an approach where prediction integrates data from various predictive variables, effectively mitigating the shortcomings associated with reliance on a single attribute.

Further experimental analyses were conducted to elucidate the role of structural features predicted by AlphaFold2 (Category D) in enhancing our model. These studies involved integrating structural features into existing datasets and assessing their impact on model performance. Improvements in all performance indicators were observed when these features were combined with features A, B, and C (A+D, B+D, and C+D) (Table 3). Specifically, the integration of Feature D with Feature A (A+D) was associated with an increase in classification accuracy from 84.36% to 85.43%, a relative improvement of approximately 1.27%; the AUC value was observed to rise from 0.9167 to 0.9313, marking a relative increase of about 1.52%. Similarly, the combination of Feature D with Feature B (B+D) was found to enhance classification accuracy from 88.68% to 89.02%, a relative increase of approximately 0.34%; the AUC value improved from 0.9474 to 0.9552, a relative increase of about 0.82%. Figure 3 graphically represents the enhancements achieved by integrating Feature D with Feature C in our predictive model. It visually demonstrates the improvement in classification accuracy, from 90.48% to 91.07%, and the increase in the AUC value, from 0.9625 to 0.9654. These graphical depictions emphasize the impactful role of AlphaFold2’s structural features in enhancing model performance, particularly in terms of accuracy and AUC, underscoring their utility in predicting missense mutations.

To explore the impact of various feature combinations on model performance, a series of incremental feature combination experiments were undertaken. These experiments began with basic sequence features (Category A) and progressively integrated hybrid structure and function features (Category B), integrated predictive tool features (Category C), and ultimately AlphaFold2 structural features (Category D). 

Figure 4 is designed to clearly illustrate the progressive impact of incorporating diverse feature types on enhancing all key performance metrics of our predictive model. According to the results presented in Table 4, when only Category A features are utilized for predictions, a classification accuracy of 84.36% and an AUC of 0.9167 are achieved by the model. With the incorporation of Category B features (A + B), significant enhancements in model performance are observed: accuracy is elevated to 88.86%, marking a 5.34% increase relative to the usage of Category A features alone; AUC ascends to 0.9528, reflecting a 3.94% improvement. The subsequent addition of Category C features (A + B + C) further elevates the assessment metrics: accuracy rises to 91.26%, a 2.70% gain over the A + B configuration; AUC climbs to 0.9676, a 1.55% enhancement. Ultimately, the inclusion of all features (A + B + C + D) results in optimal performance: accuracy reaches 91.82%, a 0.61% increment from the A + B + C setup; AUC advances to 0.9701, a 0.26% increase.

### 6.2. Comparison with Commonly Used Machine Learning Models

In previous research, pathogenicity predictions for missense variants were primarily based on traditional machine-learning techniques that utilized manually extracted features along with established bioinformatics parameters, such as amino acid conservation, physicochemical properties, and gene functional domains (as seen in tools like PolyPhen-2 and SIFT). Due to their robust predictive accuracy, algorithms such as Random Forest (RF) [57], Logistic Regression (LR) [58], and Support Vector Machines (SVM) [59] have been widely adopted in classification tasks. In this study, a MissenseNet was constructed, and the chosen machine learning algorithms were subjected to tenfold cross-validation using the previously described feature combination (A + B + C + D). The evaluation scores obtained are presented in the results section.

Compared to RF and LR, the newly developed MissenseNet exhibited enhanced performance across various metrics, including accuracy, recall, precision, F1 score, Matthews Correlation Coefficient (MCC), balanced accuracy, AUC, and AUC-PR. Figure 5 provides an in-depth visualization of the experimental outcomes, specifically highlighting the superior performance of MissenseNet in various comparative metrics. This figure presents a detailed graphical comparison between MissenseNet and other established methods like Random Forest (RF) and Logistic Regression (LR) across key performance indicators such as recall, Matthew’s Correlation Coefficient (MCC), Area Under the Curve (AUC), and Precision-Recall AUC (AUC-PR). It effectively showcases MissenseNet’s top accuracy rate of 91.82% and its remarkable AUC of 0.9701, as shown in Table 5, visually demonstrating its predictive accuracy and reliability. Through this figure, we aim to clearly illustrate MissenseNet’s standout performance, affirming its significant potential for clinical application and its reliability in predicting the pathogenicity of missense variants.

### 6.3. Comparison with Typical Deep Learning Models

To further explore the capabilities of MissenseNet, this study conducted comparative analyses with several leading deep learning-based classification models, including MLP [60], RNN, CNN, DenseNet, and ResNet. These models were selected due to their widespread use and proven efficacy across a variety of machine-learning tasks in recent years. In this study, each model, incorporating the previously described feature combination (A + B + C + D), underwent multiple iterations of tenfold cross-validation on our dataset. Critical performance metrics—including accuracy, recall, precision, F1 score, Matthews correlation coefficient (MCC), balanced accuracy, and the areas under the receiver operating characteristic curve (AUC) and the precision-recall curve (AUC-PR)—were rigorously evaluated.

Notably, MissenseNet achieved an accuracy of 91.82%, significantly surpassing that of other benchmark models, with CNN being the closest competitor at 90.61%. Furthermore, MissenseNet excelled in the recall at 87.76% and maintained competitive precision at 90.20%, effectively identifying true positives without significantly increasing false positives. With an MCC of 82.53%, MissenseNet delivered superior predictions, vastly outperforming simpler models such as RNN and closely following CNN. It also led to balanced accuracy and both AUC metrics (ROC and PR), achieving 97.01% and 95.49%, respectively, as detailed in Table 6. These robust performances not only confirm the model’s superior operational traits across various thresholds but also underscore its utility in scenarios where the accuracy of positive and negative predictions carries significant implications. Figure 6 offers a comprehensive visualization of the experimental results, clearly showcasing MissenseNet’s outstanding performance relative to other models. The figure uses a radar chart to compare the performance of MissenseNet against competing models across a variety of metrics. In this visual representation, MissenseNet is consistently positioned on the outermost ring, indicating its superior performance in metrics such as accuracy, precision, recall, and F1-score.

### 6.4. Results of Ablation Experiments with SE and Encoder-Decoder Modules

To validate the contributions of each enhancement proposed in this study to network performance, the proposed improvements were sequentially integrated into the model for ablation studies. The experiments involved the incorporation of the SE module and an encoder-decoder module. Following exhaustive testing and cross-validation using a consistent dataset, the results are shown in Table 7.

Table 7 clearly illustrates that the baseline model, MissenseNet, was observed to achieve an accuracy of 89.81% with an AUC of 96.83. With the integration of the encoder-decoder module, an increase in accuracy to 91.67% was recorded, representing a relative improvement of 2.07% over the baseline MissenseNet, and the AUC was elevated to 0.9700. Further incorporation of the SE module resulted in the accuracy being raised to 91.82% and the AUC to 0.9701, constituting a relative improvement of 2.24% compared to the baseline. These results not only confirm the notable enhancements yielded by the modifications relative to the baseline model but also underscore the precision with which the model processes complex datasets. Designed to inherit the efficient features of ShuffleNet, MissenseNet markedly enhances performance and generalization capabilities through innovative network structures and training techniques, highlighting its potential for real-world applications, particularly in contexts demanding high accuracy and reliability.

### 6.5. Comparison with Single-Type Forecasting Tools

In the independent test set, the performance of MissenseNet, a model developed in this study, was evaluated against single-predictor tools. The Receiver Operating Characteristic (ROC) and Precision-Recall (PR) curves, as shown in Figure 7 and Figure 8, clearly demonstrate that MissenseNet outperforms the other tools. The ROC curve of MissenseNet nearly reaches the upper left corner of the graph, indicating a high true positive rate alongside a low false positive rate, thereby marking substantial improvements over competing tools, particularly in terms of maximizing true positives and minimizing false positives. These results highlight the reliability and accuracy of MissenseNet in assessing the impacts of single nucleotide variants (SNVs).

Furthermore, MissenseNet was compared to other single-predictor tools using the Precision-Recall Curve (PR Curve), which assesses the model’s precision at various levels of recall and is particularly valuable for datasets with class imbalances. Ideally, a PR curve should closely approach the graph’s upper right corner, signifying both high precision and recall. The displayed PR curve shows that MissenseNet consistently maintained high precision across all levels of recall, achieving an area under the average precision curve (APUC) of 0.9491. This performance indicates that MissenseNet exhibits exceptional predictive accuracy for positive samples across various threshold settings.

### 6.6. Comparison with Integrated Forecasting Tools

Figure 9 showcases the ROC curve analysis of MissenseNet in comparison with various leading predictive models, emphasizing its superior performance in the field. The ROC curve of MissenseNet nearly reaches the upper left corner, indicating a high true positive rate coupled with a low false positive rate, with an AUC of 0.9662, which highlights its superior classification capability. When compared with advanced comprehensive predictive tools such as REVEL, VEST4, MetaLR, and DEOGEN2—all of which exhibited strong predictive abilities with AUC values above 0.90—MissenseNet still achieved a higher AUC. It was notably close to REVEL and VEST4, which recorded AUCs of 0.9554 and 0.9505, respectively. MissenseNet’s curve, approaching the upper left corner, illustrates its efficacy in maintaining a low false positive rate while securing a higher true positive rate. This nuanced yet significant difference accentuates MissenseNet’s capability to accurately differentiate between positive and negative classes. Against tools like CADD, MVP, and others, MissenseNet’s AUC was significantly higher. For tools with lower AUC values, such as DANN and GenoCanyon, the performance advantage of MissenseNet was even more marked, further establishing its reliability and effectiveness as a comprehensive predictive tool, especially in complex scenarios that require precise distinctions. Overall, comprehensive evaluation tools typically surpass single predictor tools, and MissenseNet achieved the best classification outcomes among all models compared.

As shown in Figure 10, MissenseNet’s PR curve is closer to the upper right corner, reflecting its exceptional precision and recall across all levels. With an APUC value of 0.9491, MissenseNet significantly outperformed other comprehensive predictive tools like REVEL, MetaSVM, and MetaLR, which had APUC values of 0.9332, 0.8772, and 0.8874, respectively. Compared to these tools, MissenseNet exhibited greater precision at higher recall levels, underscoring its accuracy and reliability in identifying missense variants.

## 7. Conclusions

In genetic disease research, the accurate prediction of missense variant pathogenicity is crucial for understanding the molecular bases of diseases and advancing personalized medicine. Although numerous prediction methods exist, they often rely solely on basic genetic data, standard bioinformatics tools, or single-algorithm models, and may not effectively capture the complex genetic backgrounds or the subtle relationships between variants and diseases. To address these limitations, MissenseNet, a novel deep learning-based model, has been developed. This model integrates comprehensive genetic and biological data to improve the accuracy of pathogenicity assessments for missense variants. Designed for computational efficiency and lightness, MissenseNet ensures robust performance on receiver operating characteristic (ROC) and precision-recall (PR) curves in large datasets.

In the realm of genetic disease prediction, missense variants located in the core regions of protein tertiary structures are found to be more likely to be associated with diseases than those on the surface. To augment the model’s performance, sequence-based features (Category A), hybrid structural and functional information (Category B), comprehensive features (Category C), and newly utilized structural features generated by AlphaFold2 (Category D) have been integrated. AlphaFold2, renowned for its high predictive accuracy and biological relevance, is anticipated to significantly enhance model performance through the integration of these structural features. The performance metrics under various model configurations, including accuracy, recall, precision, and F1 scores, have been meticulously recorded and compared, assessing the significance of improvements through statistical analysis. Ablation experiments have indicated that the inclusion of Category D features into Categories A, B, and C individually boosts model performance. Notably, the addition of Category D features to Category A alone has improved the AUC by 1.52%. When combining Categories A, B, and C with Category D features, the average AUC of the model has increased by 0.26%. These results highlight the importance of incorporating a complete feature set (A + B + C + D) in the predictive model to achieve optimal prediction outcomes.

In this study, MissenseNet, a deep learning algorithm based on the ShuffleNet architecture, was developed to enhance the prediction of pathogenicity in missense variants. Known for its efficient parameter utilization and computational performance, ShuffleNet is particularly suited for processing large-scale bioinformatics data. Strategic enhancements in ShuffleNet, designed to optimize performance within computational constraints, have further refined the model. These advancements are suggested to improve the model’s capabilities. Integrating the SE module (Squeeze-and-Excitation Module) and an encoder-decoder network structure has significantly enhanced the model’s feature processing and utilization capabilities. Feature channel dependencies are strengthened, and the features are effectively recalibrated by the SE module, enhancing the focus on pertinent features. Concurrently, the effective fusion of high-level and low-level features is promoted by the encoder-decoder structure, optimizing information flow and thus improving the interpretability and precision of outputs. Refinements in ShuffleNet have also been instrumental in enhancing the reliability of MissenseNet, which leverages this architecture to predict the pathogenicity of missense variants. This is particularly beneficial in supporting clinical decision-making processes. Validated through evaluations on an independent test set, the performance of MissenseNet, particularly excelling in the area under the Receiver Operating Characteristic (ROC) curve and Precision-Recall (PR) curve with an AUC of 0.9662, demonstrates not only high accuracy in classification tasks but also excellent generalization ability. Compared to existing prediction tools, a significant enhancement in performance in distinguishing disease-associated variants from benign variants by MissenseNet has been noted. These improvements underscore that the incorporation of the SE module and encoder-decoder network into the ShuffleNet architecture has not only bolstered the model’s feature expression and processing capabilities but also markedly increased the accuracy of pathogenicity predictions for missense variants.

While the MissenseNet model has enhanced the prediction of missense variants, significant potential remains to further improve its accuracy and generalizability. The effectiveness of the Class D structural features used in this research depends critically on the precision of the AlphaFold2 structural model. Although AlphaFold2 is adept at predicting the overall protein structure, its limitations become evident when addressing intrinsically disordered proteins (IDPs) and regions with minimal structure. These proteins and regions lack a stable tertiary structure, and AlphaFold2, primarily trained on well-structured protein data, may struggle to accurately predict these structures. This is particularly crucial, as IDPs play key roles in cellular functions and are linked to various diseases. Additionally, AlphaFold2 may not accurately capture subtle structural changes induced by specific genetic variants. A small-scale study has shown that using AlphaFold to evaluate conformational shifts between wild-type and variant forms does not precisely determine the impacts of these variations. Moreover, AlphaFold2 has not yet been able to fully replace experimental validation, and evidence suggests that it sometimes inaccurately predicts the structural effects of variants known to cause structural alterations [61]. An over-reliance on data predicted by AlphaFold2 or similar tools without sufficient experimental confirmation may lead to theoretical assessments of the pathogenicity of missense variants that overlook the inherent biological complexity and context dependency.

Faced with complex and variable genetic data, the effectiveness of deep learning algorithms in enhancing both the accuracy and efficiency of pathogenicity predictions for missense variants has been confirmed by our research. As AlphaFold3 [62] emerges, we plan to explore its enhanced capabilities to further enrich our methodology. Notably, AlphaFold3 offers improved prediction accuracy, particularly for intrinsically disordered proteins and regions with minimal structural data, addressing some limitations of its predecessor. Additionally, its advanced machine learning techniques could provide more nuanced insights into subtle genetic variations impacting protein structures. Future endeavors could draw on successful strategies from areas such as image recognition and natural language processing, applying transfer learning, and multi-task learning to capitalize on knowledge gained in other bioinformatics tasks. This approach would enhance the model’s capacity to process data from new domains. By incorporating interpretative techniques such as feature importance analysis, visualization technologies, and reverse engineering tools, the transparency of the model’s decision-making processes can be bolstered, thereby increasing researchers’ confidence in the predictions. This integration of AlphaFold3 will allow for more precise and robust models, potentially revolutionizing our understanding of genetic pathogenicity in complex diseases.

In summary, MissenseNet is an innovative tool specifically designed to predict the pathogenicity of missense variants, integrating advanced structural features from AlphaFold2 with deep learning algorithms. This method not only improves the accuracy of predictions but also deepens our understanding of the pathogenic mechanisms of missense variants through detailed analysis of protein structure changes. Having been experimentally validated, its interpretative capabilities surpass those of many traditional prediction tools. Looking ahead, plans are in place to integrate more complex data sources and develop more sophisticated algorithms to continuously improve and refine our predictive capabilities for missense variants. Through these advancements, MissenseNet is poised to become an essential tool in genetic disease research and precision medicine.

## Figures and Tables

**Figure 1 biomolecules-14-01105-f001:**
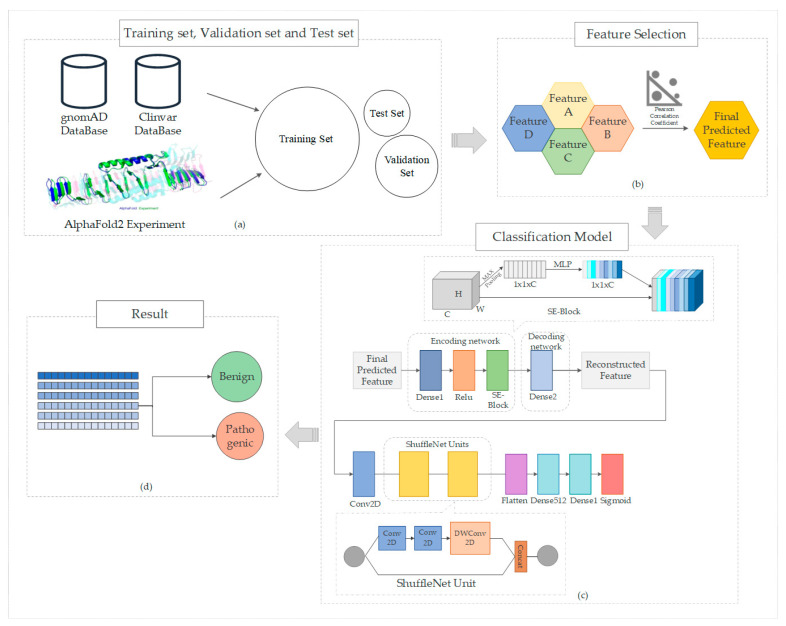
Comprehensive workflow of this study: (**a**) data collection, where datasets pertinent to our study are gathered, including the extraction of missense variant information from the gnomAD and ClinVar databases, complemented by structural features predicted using AlphaFold2; (**b**) feature selection, highlighting the criteria and methods used to identify relevant features for analysis; (**c**) model construction, detailing the development and architecture of our predictive model. Our model is based on the ShuffleNet unit, enhanced with an encoder-decoder network structure and the Squeeze-and-Excitation (SE) module; (**d**) prediction outcomes, showcasing the results obtained from applying our model, specifically categorizing variants as pathogenic or benign.

**Figure 2 biomolecules-14-01105-f002:**
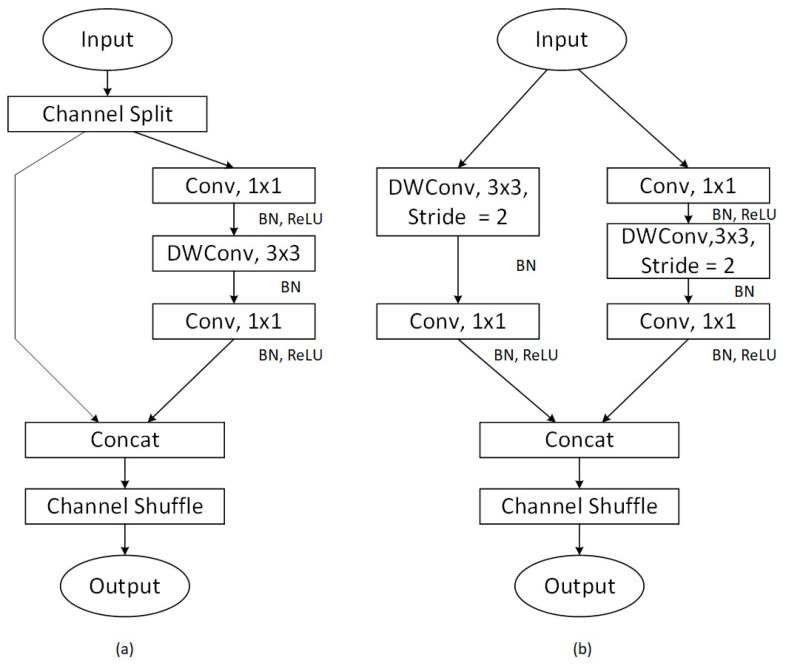
ShuffleNet v2 basic architecture diagram (**a**) the basic ShuffleNet V2 unit; (**b**) ShuffleNet V2 unit for spatial down sampling.

**Figure 3 biomolecules-14-01105-f003:**
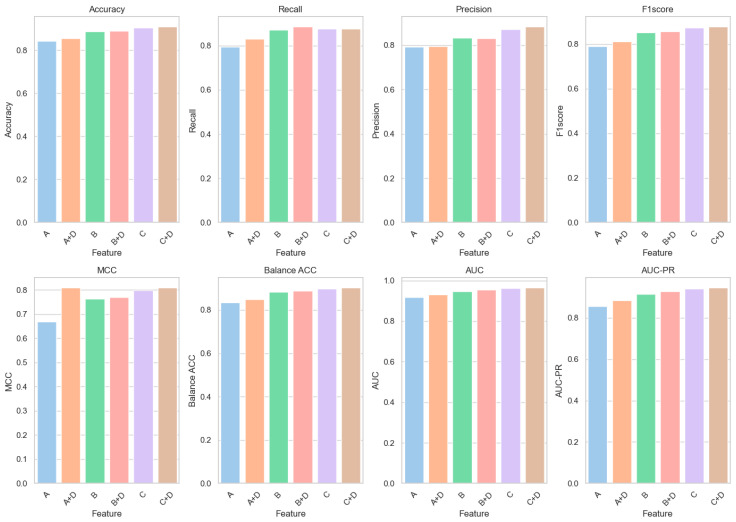
Basic feature fusion D-type feature result diagram.

**Figure 4 biomolecules-14-01105-f004:**
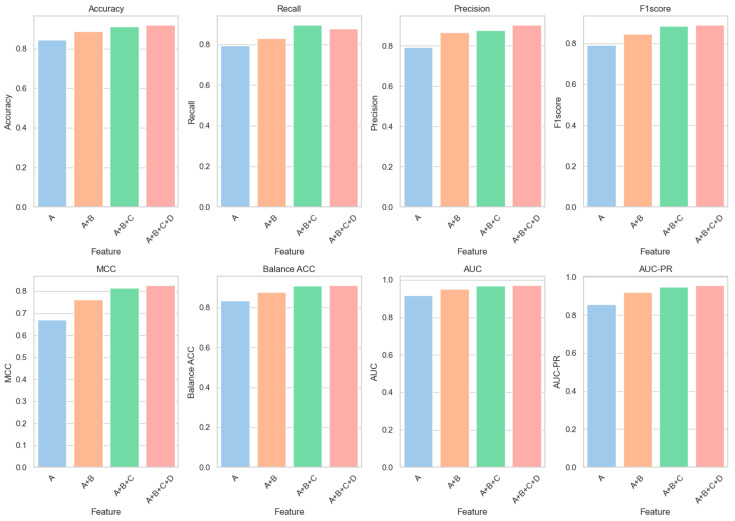
Feature fusion result diagram.

**Figure 5 biomolecules-14-01105-f005:**
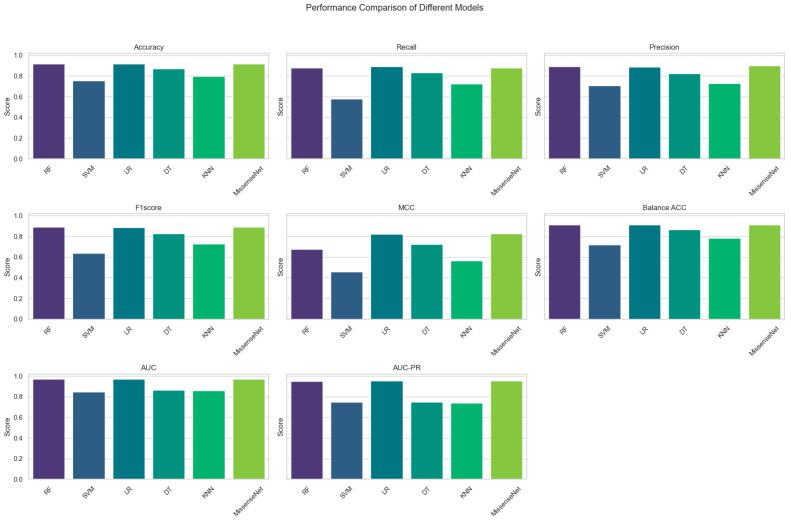
Comparison results of MissenseNet with commonly used machine learning.

**Figure 6 biomolecules-14-01105-f006:**
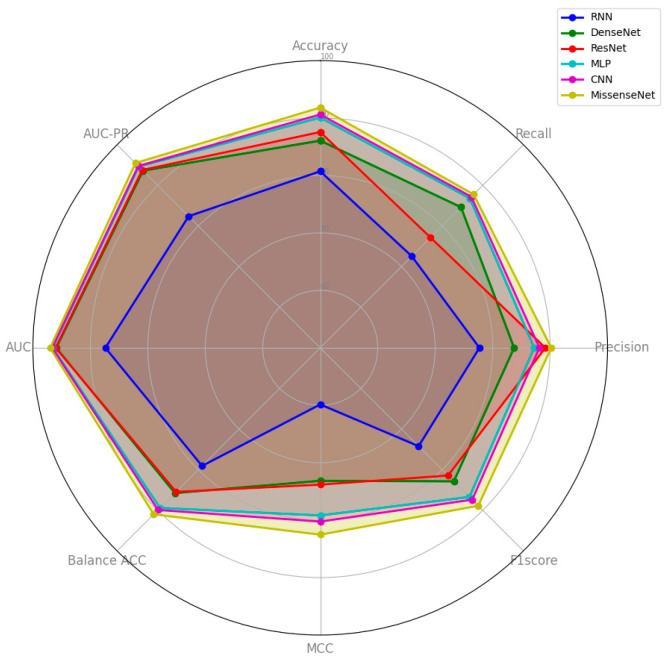
Comparison results between MissenseNet and typical deep learning.

**Figure 7 biomolecules-14-01105-f007:**
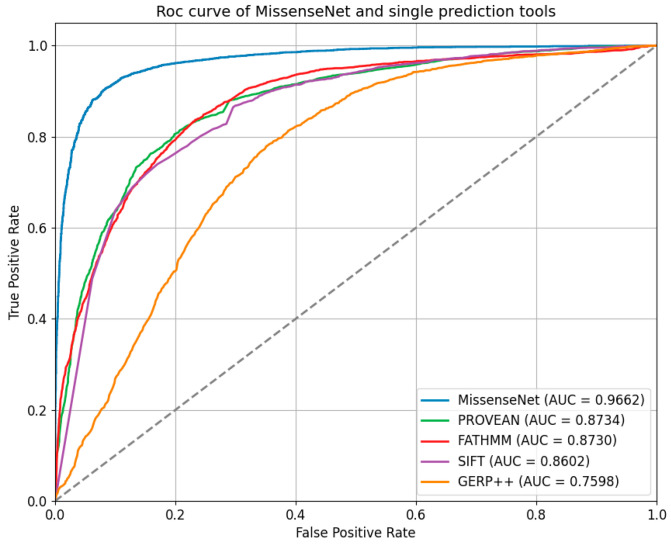
Roc curve of MissenseNet and single prediction tools.

**Figure 8 biomolecules-14-01105-f008:**
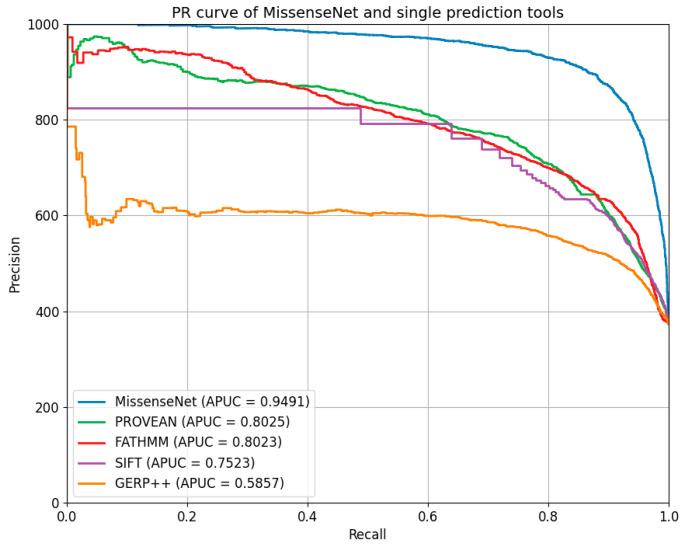
PR curve of MissenseNet and single prediction tools.

**Figure 9 biomolecules-14-01105-f009:**
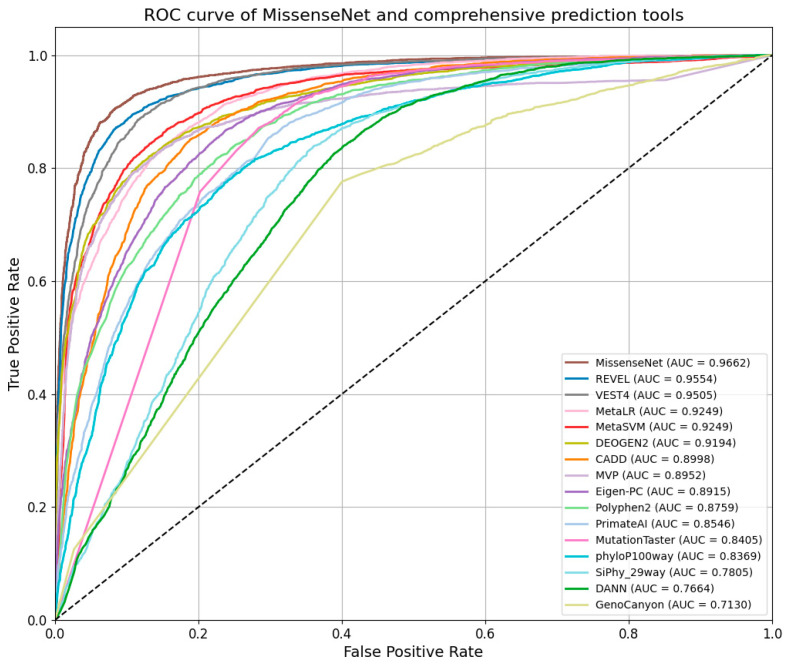
Roc curve of MissenseNet and comprehensive prediction tools.

**Figure 10 biomolecules-14-01105-f010:**
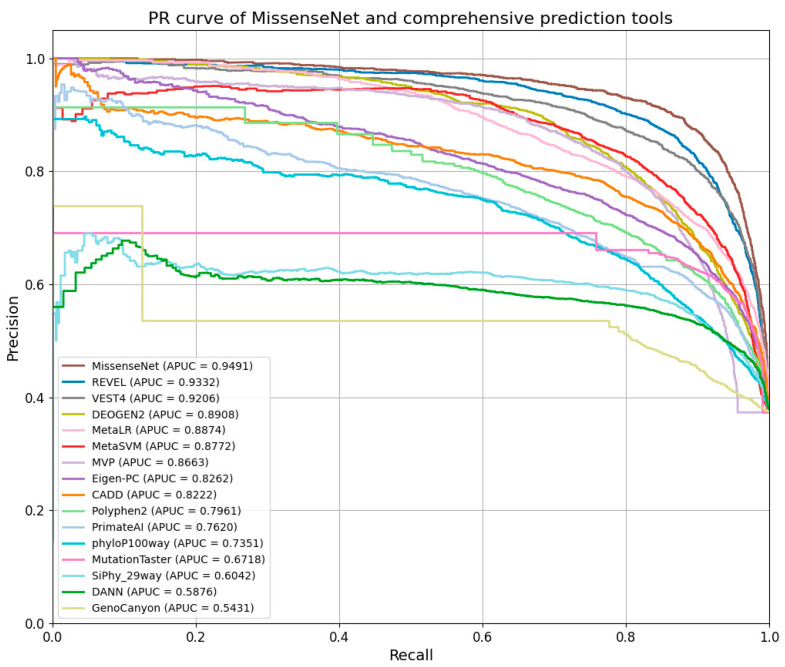
PR curve of MissenseNet and comprehensive prediction tools.

**Table 1 biomolecules-14-01105-t001:** Common pathogenicity prediction tools.

Single-type forecasting tool	Sequence-based prediction methods (A)	SIFT [18]
		SIFT4GSIF [28]
		PROVEAN [27]
		GERP++ [30]
		phyloP100way [31]
		phyloP30way [32]
		SiPhy [33]
	Methods integrating structural and functional features (B)	Polyphen2_HDIV [14]
		Polyphen2_HVAR [14]
		MutationTaster2 [34]
		VEST4 [35]
		Fathmm [36]
		GenoCanyon [37]
29Ensembletools (C)	REVEL [38]	
	CADD [39]	
	DANN [40]	
	MetaSVM [41]	
	MetaLR [41]	
	DEOGEN2 [42]	
	PrimateAI [43]	
	Eigen [44]	

**Table 2 biomolecules-14-01105-t002:** Four-category classification results of basic features.

Feature	Accuracy	Recall	Precision	F1 Score	MCC	Balance ACC	AUC	AUC-PR
A	0.8436	0.7938	0.7930	0.7905	0.6690	0.8335	0.9167	0.8554
B	0.8868	0.8717	0.8343	0.8516	0.7618	0.8837	0.9474	0.9148
C	0.9048	0.8762	0.8715	0.8727	0.7980	0.8990	0.9625	0.9411
D	0.7053	0.4943	0.6736	0.5302	0.3523	0.6626	0.7834	0.6806

**Table 3 biomolecules-14-01105-t003:** Basic feature fusion D-class feature results.

Feature	Accuracy	Recall	Precision	F1 Score	MCC	Balance ACC	AUC	AUC-PR
A + D	0.8543	0.8306	0.7940	0.8121	0.8095	0.8495	0.9306	0.8831
B + D	0.8898	0.8867	0.8305	0.8571	0.7694	0.8892	0.9552	0.9265
C + D	0.9101	0.8773	0.8841	0.8792	0.8095	0.9035	0.9655	0.9461

**Table 4 biomolecules-14-01105-t004:** Feature fusion results.

Feature	Accuracy	Recall	Precision	F1 Score	MCC	Balance ACC	AUC	AUC-PR
A	0.8436	0.7938	0.7930	0.7905	0.6690	0.8335	0.9167	0.8554
A + B	0.8886	0.8306	0.8668	0.8471	0.7613	0.8768	0.9528	0.9191
A + B + C	0.9126	0.8945	0.8762	0.8843	0.8154	0.9089	0.9676	0.9492
A + B + C + D	0.9182	0.8776	0.9020	0.8889	0.8253	0.9100	0.9701	0.9549

**Table 5 biomolecules-14-01105-t005:** Comparison results of MissenseNet with commonly used machine learning.

Model	Accuracy	Recall	Precision	F1 Score	MCC	Balance ACC	AUC	AUC-PR
RF	0.9178	0.8804	0.8916	0.8896	0.6741	0.9118	0.9686	0.9506
SVM	0.7528	0.5807	0.7055	0.6368	0.4573	0.7181	08465	0.7480
LR	0.9160	0.8899	0.8861	0.8877	0.8209	0.9107	0.9692	0.9523
DT	0.8705	0.8301	0.8245	0.8271	0.7238	0.8623	0.8623	0.7479
KNN	0.7954	0.7245	0.7269	0.7255	0.5628	0.7811	0.8586	0.7401
MissenseNet	0.9182	0.8776	0.9020	0.8889	0.8253	0.9100	0.9701	0.9549

**Table 6 biomolecules-14-01105-t006:** Comparison results between MissenseNet and typical deep learning.

Model	Accuracy	Recall	Precision	F1 Score	MCC	Balance ACC	AUC	AUC-PR
MLP	0.9009	0.8682	0.8724	0.8673	0.7919	0.8943	0.9658	0.9466
RNN	0.8073	0.7254	0.7771	0.7422	0.5986	0.7908	0.8739	0.8239
CNN	0.9061	0.8719	0.8815	0.8740	0.8025	0.8992	0.9666	0.9477
DenseNet	0.8607	0.8463	0.8374	0.8288	0.7316	0.8577	0.9594	0.9361
ResNet	0.8756	0.7715	0.8918	0.8145	0.7382	0.8546	0.9605	0.9379
MissenseNet	0.9182	0.8776	0.9020	0.8889	0.8253	0.9100	0.9701	0.9549

**Table 7 biomolecules-14-01105-t007:** Ablation Experiment Results.

Model	Factor		Acc	Recall	Precision	F1 Score	AUC	AUC-PR
	SE Module	Encoder-Decoder Module						
MissenseNet (Baseline)	/	/	0.8981	0.8459	0.8892	0.8606	0.9683	0.9512
MissenseNet-(EN)	/	√	0.9167	0.8714	0.9034	0.8864	0.9700	0.9540
MissenseNet (SE + EN)	√	√	0.9182	0.8776	0.9020	0.8889	0.9701	0.9549

Note: ‘√’ indicates inclusion of the module, ‘/’ indicates exclusion.

## Data Availability

Data are available from the corresponding author.

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
