# Peer review of "Enhancing Missense Variant Pathogenicity Prediction with MissenseNet: Integrating Structural Insights and ShuffleNet-Based Deep Learning Techniques"

_biomolecules, 2024, doi:10.3390/biom14091105_

Round 1

Reviewer 1 Report

Comments and Suggestions for Authors

In this paper titled “Enhancing Missense Variant Pathogenicity Prediction with MissenseNet: Integrating Structural Insights and ShuffleNet-Based Deep Learning Techniques” the authors mentioned about the efficacy of their algorithm to identify the disease variants and their classification for pathogenicity were validated with superior accuracy over the conventional methods. They have introduced MissenseNet, a model has advanced beyond standard predictive features, incorporates structural insights from AlphaFold2 protein predictions, thus optimizing structural data utilization. In this model, they have incorporated an encoder-decoder framework and a Squeeze-and-Excitation (SE) module on the ShuffleNet-Based Deep Learning Techniques. The whole work is very analytical and well written but there are few suggestions may help the authors to make a better paper for the wide audience.

1) In the abstract, authors used four abbreviations (ROC, PR, AUC and APUC). Please write the full form as well, even though they are very familiar abbreviations.

2) References are required for (Page 1 and line 43) “they do not consistently yield reliable predictions regarding the pathogenicity of mutations.”

3) Reference is required for “PolyPhen-2 was a pioneer in incorporating such data, utilizing metrics such as residue accessibility, shifts in hydrophobic tendencies, and crystallographic B factors for proteins with available structures.” Otherwise, it looks like a random statement.

4) Reference is required for “machine learning models often require a clearly defined problem framework and specific input features, which may not always be feasible due to the variability inherent in biological diversity.”

5) Author may avoid “Figure 1 illustrates the overall structure of the paper.” like sentences. They may frame the sentence in different form.

6) Caption of the Figure 1 is very short and requires more explanation.

7) I am just wondering if author widely used AlphaFold2 for their work, why did they ignore the main nature paper of AlphaFold2 (Jumper et al Nature 596pages583–589 (2021))?

8) All the captions of the figures are not well explained, it needs serious attention.

9) Figure 9 and 10, it needs some modification. The offset models are not visible clearly. Authors may exclude the less important models and place it in the supplementary data. Most important models may be included in the main text.

Major issues

1) How do the authors defend the application of AlphaFold2 in the era of AlphaFold3 for their problem?

2) The authors did not mention the limitation of the AlphaFold2 for the less structured proteins, intrinsically disordered proteins. The confidence level is >50%. This limitation will be translated to their model if it heavily depends on AlphaFold2. The author may highlight and comment on the remedies of this problem.

3) When the authors claim about the pathogenicity of the disease variants, the sequence to structural correlation is understandable but the functional part connection is missing. Only model validation was done on the literature survey and prior computational report, or some novel and unique variants that their model determines different from the preceding model, any validation work were done on these variants?

4) In my opinion, the importance or advantage of MissenseNet is missing. The limitations were also mentioned in scattered ways.    

Author Response

Comments and Suggestions for Authors
1:

In this paper titled “Enhancing Missense Variant Pathogenicity Prediction with MissenseNet: Integrating Structural Insights and ShuffleNet-Based Deep Learning Techniques” the authors mentioned about the efficacy of their algorithm to identify the disease variants and their classification for pathogenicity were validated with superior accuracy over the conventional methods. They have introduced MissenseNet, a model has advanced beyond standard predictive features, incorporates structural insights from AlphaFold2 protein predictions, thus optimizing structural data utilization. In this model, they have incorporated an encoder-decoder framework and a Squeeze-and-Excitation (SE) module on the ShuffleNet-Based Deep Learning Techniques. The whole work is very analytical and well written but there are few suggestions may help the authors to make a better paper for the wide audience.

Question 1:In the abstract, authors used four abbreviations (ROC, PR, AUC and APUC). Please write the full form as well, even though they are very familiar abbreviations.

Response:

Dear Reviewer,

Thank you for your meticulous review and valuable suggestions. Regarding your comment about the four abbreviations used in the abstract (ROC, PR, AUC, and APUC), we understand that even though these terms are commonly understood within the field, clarity and readability of the manuscript demand that their full forms be provided when they first appear. We will make the following corrections in the revised manuscript:

ROC: Receiver Operating Characteristic

PR: Precision-Recall

AUC: Area Under the Curve

APUC: Area Under the Precision-Recall Curve

We will ensure that these terms are presented in their full forms at their first occurrence in the abstract to aid all readers, including those outside the specialized field, in better understanding their meanings.

Thank you again for your guidance. We look forward to your further feedback on the revised manuscript.

Question 2:References are required for (Page 1 and line 43) “they do not consistently yield reliable predictions regarding the pathogenicity of mutations.”

Response:

Dear Reviewer,

Thank you for your feedback and guidance. Regarding the statement on page 1, line 43, "They do not always provide consistent and reliable predictions regarding the pathogenicity of mutations1," we indeed need to add relevant references to support this view.

Accordingly, we will cite the following publication:

Ge Fang, Hu Jun, Zhu Yiheng, Yu Dongjun. Review of Predictive Studies on the Pathogenicity of Non-synonymous SNVs. Journal of Nanjing University of Science and Technology, 2021, 45(01):1-17.

This literature review addresses the current state and challenges of predicting the pathogenicity of non-synonymous single nucleotide variants (nsSNVs), highlighting the limitations in the consistency and reliability of existing methods.

We will add this reference to our paper to more accurately reflect the current understanding and research progress in the scientific community regarding this issue. Thank you again for your meticulous review and valuable suggestions. We look forward to your further feedback on the revised manuscript.

Question 3:Reference is required for “PolyPhen-2 was a pioneer in incorporating such data, utilizing metrics such as residue accessibility, shifts in hydrophobic tendencies, and crystallographic B factors for proteins with available structures.” Otherwise, it looks like a random statement.

Response:

Dear Reviewer,

Thank you for your attention to the accuracy of citations in our manuscript. Regarding the statement that "PolyPhen-2 was a pioneer in utilizing such data, employing metrics such as residue accessibility, shifts in hydrophobic tendencies, and crystallographic B factors for proteins with available structures,2" we recognize the need to provide specific references to support this viewpoint.

This statement is derived from PolyPhen-2's advanced method of integrating various biophysical and biochemical parameters for predicting the pathogenicity of missense mutations. To offer clear bibliographic support, we refer to the following key publication:

Adzhubei IA, Schmidt S, Peshkin L, et al. A method and server for predicting damaging missense mutations. Nat Methods. 2010;7(4):248-249.

This publication details how PolyPhen-2 utilizes features such as residue accessibility, changes in hydrophobic tendencies, and crystallographic B factors to predict the pathogenicity of missense mutations, providing scientific basis for our description.

We will ensure this reference is explicitly cited in the revised manuscript and that all crucial statements are supported by reliable sources. Thank you again for your suggestions and guidance. We look forward to your further feedback on the revised manuscript.

Question 4:Reference is required for “machine learning models often require a clearly defined problem framework and specific input features, which may not always be feasible due to the variability inherent in biological diversity.”

Response:

Dear Reviewer,

Thank you for your request for references to support the statement on the challenges of defining problem frameworks and specific input features for machine learning models in the context of biological diversity. The variability inherent in biological data can indeed complicate the application of machine learning techniques, which often require precise and consistent input data.

We have referenced two relevant articles that discuss these challenges and the broader implications for machine learning in bioinformatics:

Min, S., Lee, B., & Yoon, S. (2017). "Deep learning in bioinformatics." Briefings in Bioinformatics, 18(5), 851-869. DOI: 10.1093/bib/bbw068. This article provides an overview of how deep learning techniques are applied in bioinformatics, addressing the complexities and variability in biological data.

Angermueller, C., Pärnamaa, T., Parts, L., & Stegle, O. (2016). "Deep learning for computational biology." Molecular Systems Biology, 12(7), 878. DOI: 10.15252/msb.20156651. This publication discusses the adaptation of deep learning methods to the unique challenges posed by the variability and complexity of biological systems.

These references are integral to understanding the specific demands and limitations of applying machine learning methodologies in the field of bioinformatics and will be cited appropriately in our manuscript to substantiate the discussed points.

We hope this addresses your query satisfactorily and strengthens the manuscript by providing a solid reference framework for the challenges mentioned.

Question 5:Author may avoid “Figure 1 illustrates the overall structure of the paper.” like sentences. They may frame the sentence in different form.

Response:

Dear Reviewer,

Thank you for your constructive feedback regarding the style of expression used in our manuscript. We acknowledge the need to adopt a more formal and scholarly tone throughout our document. In light of your suggestions, we will revise not only the specific statement concerning Figure 1 but also similar expressions throughout the manuscript to ensure they align with academic standards.

In response to your suggestion, we have revised the sentence to reflect a more passive and formal style. The revised sentence is as follows: "The overall structure of the paper is depicted in Figure 1." This modification aims to maintain objectivity and align with the scholarly tone expected in academic publications.

We hope this revision meets your expectations and enhances the clarity and professionalism of our manuscript. We are committed to improving our work based on your recommendations and look forward to any further suggestions you might have.

Question 6:Caption of the Figure 1 is very short and requires more explanation.

Response:

Dear Reviewer,

Thank you for your feedback regarding the caption of Figure 1. We acknowledge that the initial caption was too concise and may not have provided sufficient explanation to fully understand the figure's content and its relevance to our study. We have revised the caption to include a more detailed description of what the figure represents.

We believe that this enhancement will make the figure more informative and accessible to readers, thereby improving the overall quality and clarity of our paper.

Thank you once again for your constructive comment, which has helped us improve the manuscript.

Question 7:I am just wondering if author widely used AlphaFold2 for their work, why did they ignore the main nature paper of AlphaFold2 (Jumper et al Nature 596, pages583–589 (2021))?

Response:

Dear Reviewer,

Thank you for your observation regarding the citation of the seminal Nature paper on AlphaFold2 by Jumper et al. (Nature 596, pages 583–589, 2021). We indeed utilize AlphaFold2 extensively in our work and have acknowledged this pivotal publication. The reference to this paper was included at the first instance where AlphaFold2 is mentioned in our manuscript; it is listed as reference number 15. We apologize for any oversight that may have made this citation less prominent. We appreciate your attention to detail and will ensure that the reference is more clearly highlighted in our revised manuscript to emphasize the importance of this foundational work.

Thank you once again for bringing this to our attention.

Question 8:All the captions of the figures are not well explained, it needs serious attention.

Response:

Dear Reviewer,

Thank you for your feedback regarding the explanations associated with the figures in our manuscript, both within the text and in the figure captions. We appreciate your comprehensive review and understand that both elements are crucial for clearly conveying the intended messages of our visual data.

We acknowledge that our manuscript may have fallen short in fully integrating the figures into the narrative of the text and in providing detailed captions that adequately describe and contextualize the figures. We understand that the captions should not only describe what is depicted but also explain why the figures are important and how they relate to the research questions and findings discussed in the text.

To address these concerns, we will undertake a thorough revision of both the textual references to each figure and the captions themselves. This will include:

We will expand the captions to provide more detailed descriptions, clarify the significance of the visualized data, and explain how the figures contribute to the overall research findings.

We will ensure that each figure is appropriately discussed and explained within the main body of the text. This will help to clarify any points that are visually represented and ensure that readers understand how the figures relate to the textual content.

We are committed to making these revisions meticulously to improve the clarity and effectiveness of our manuscript. We thank you again for your insights and look forward to enhancing our manuscript in line with your valuable suggestions.

Question 9:Figure 9 and 10, it needs some modification. The offset models are not visible clearly. Authors may exclude the less important models and place it in the supplementary data. Most important models may be included in the main text.

Response:

Dear Reviewer,

Thank you for your constructive feedback regarding Figures 9 and 10 in our manuscript. We appreciate your suggestions for enhancing the clarity and focus of these figures.

We acknowledge your concern that the offset models are not clearly visible. In response, we will revisit the presentation of these figures to improve their clarity and effectiveness. Our intention is to ensure that each model depicted is visible and comprehensible to all readers.

Regarding the suggestion to move less important models to supplementary data, we have carefully considered this point. We believe that all the models currently presented in Figures 9 and 10 are crucial for the comprehensive understanding of the arguments and data presented in the main text. Each model represents a critical piece of evidence supporting our findings and discussions earlier in the paper.

We will, therefore, strive to enhance the visibility and clarity of these figures without relegating any models to supplementary material. This approach will prevent potential confusion or misinterpretation that might arise from separating these models from the primary narrative of the paper.

We are committed to making the necessary adjustments to ensure that these figures meet the high standards of clarity and relevance expected in our field. We will enhance the graphical representation to make sure that the important details are clear and effectively communicated.

Thank you once again for your helpful comments. We look forward to improving our manuscript with your recommendations.

Major issues:

Question 10:How do the authors defend the application of AlphaFold2 in the era of AlphaFold3 for their problem?

Response:

Dear Reviewer,

Thank you for your attention to and suggestions for our research methodology.

Our experimental design and data collection began at the end of 2022, when AlphaFold3 3had not yet been released. Thus, the initial experimental design, simulation experiments, and subsequent data analysis were all based on AlphaFold2. Since its release, AlphaFold2 has been widely applied across multiple biological fields, and its predictions have been extensively validated for accuracy and reliability. At the start of our research, the protein structure database provided by AlphaFold2 was already comprehensive, offering a solid foundation for our study.

Although AlphaFold3 represents the latest advancement in the AlphaFold series, its performance and applicability in practical applications are still being explored.

We plan to investigate and evaluate the potential applications of AlphaFold3 in future research. The experiences from the current study will provide valuable references for using more advanced tools in our future work. We will specifically mention AlphaFold3 in the main text of our paper and explain in detail in the discussion section why this latest tool was not used in the current study and outline our plans for future research.

We understand the importance of rapid technological advancements in scientific work and appreciate your suggestions that help enhance the quality and foresight of our research. We look forward to further advancing research in related fields with the aid of new technology.

Question 11:The authors did not mention the limitation of the AlphaFold2 for the less structured proteins, intrinsically disordered proteins. The confidence level is >50%. This limitation will be translated to their model if it heavily depends on AlphaFold2. The author may highlight and comment on the remedies of this problem.

Response:

Dear Reviewer,

Thank you for highlighting the limitations of AlphaFold2, particularly its challenges with less structured and intrinsically disordered proteins. We acknowledge that the accuracy of AlphaFold2 diminishes when dealing with proteins that lack stable three-dimensional structures, which is a significant concern given our model's reliance on AlphaFold2 for structural predictions.

To address this limitation, we have enhanced our model by incorporating a variety of additional predictive features that are not solely dependent on the structural data provided by AlphaFold2. Specifically, we have integrated sequence-based features, structural and functional features, and comprehensive features, all aimed at bolstering the reliability of our predictions, especially where AlphaFold2 may fall short.

Furthermore, our model's capability has been significantly improved through the integration of the SE module (Squeeze-and-Excitation Module) and an encoder-decoder network structure. These enhancements are designed to improve our model’s feature processing and utilization capabilities, increasing adaptability to the complex nature of protein behaviors, including those of disordered proteins.

We appreciate your constructive feedback, which has prompted a more thorough evaluation of our methods. In the revised manuscript, we will ensure that these enhancements and their roles in mitigating the limitations of AlphaFold2 are clearly emphasized, ensuring transparency in our methodologies and their inherent limitations.

Additionally, we are exploring future research directions to continuously improve the accuracy and applicability of our model. Plans are in place to investigate more advanced machine learning techniques and newer versions of protein structure prediction models, potentially including AlphaFold3.

Thank you once again for your valuable insights. We are committed to refining our approach and providing a clear and comprehensive explanation of our methods and the challenges we face.

Question 12:When the authors claim about the pathogenicity of the disease variants, the sequence to structural correlation is understandable but the functional part connection is missing. Only model validation was done on the literature survey and prior computational report, or some novel and unique variants that their model determines different from the preceding model, any validation work were done on these variants?

Response:

Dear Reviewer,

Thank you for your insightful comments regarding the validation of our model, especially in terms of the connection between sequence structure and functional implications of disease variants. We appreciate the opportunity to clarify these aspects of our research.

Our research relies on datasets from widely recognized databases, namely the gnomAD population database4 (version 3.1) and the ClinVar database5 (version 20210131). These databases were not only up-to-date at the time of our study design but are also extensively used in the research community, providing a solid foundation for our analysis. The broad acceptance and use of these databases ensure the reliability of our data sources and facilitate comparability with other studies in the field.

As detailed in Table 1 of our manuscript, we have employed several methods that integrate structural and functional features to assess variant pathogenicity. These include widely utilized computational tools such as Polyphen2_HDIV2, Polyphen2_HVAR 2, MutationTaster 6, VEST4 7, Fathmm 8, and GenoCanyon 9. These tools are critical in helping to determine the potential impact of each variant, combining insights from both structural changes and functional outcomes.

We acknowledge your concern regarding the validation of novel and unique variants identified by our model as differing from previous models. Currently, our validation efforts have been focused primarily on established methodologies and comparisons with literature-reported variants. We regret not having extended our validation to these unique variants yet. Moving forward, we plan to address this gap in our future research, aiming to experimentally validate these novel predictions to solidify our model's utility and accuracy.

We hope this response addresses your concerns adequately. We are committed to continuously improving our research approach and thank you again for your valuable feedback which will undoubtedly enhance the quality of our work.

Question 13:In my opinion, the importance or advantage of MissenseNet is missing. The limitations were also mentioned in scattered ways.

Response:

Dear Reviewer,

Thank you for your valuable comments and for highlighting areas in our manuscript where the importance and advantages of MissenseNet, as well as its limitations, could be more clearly articulated. We understand your concerns and appreciate the opportunity to clarify how these elements are addressed within our text.

We have strategically placed the discussion of MissenseNet’s significance and strengths, along with its limitations, in the Conclusions section of our paper. This placement was chosen to provide a cohesive summary that ties together the findings and implications of our study, ensuring that readers can easily grasp the overarching contributions and constraints of our research in a consolidated section.

Regarding the limitations related to MissenseNet, especially those associated with its reliance on AlphaFold2, we have incorporated your mentioned limitations about AlphaFold2's challenges in handling proteins with minimal structure or that are intrinsically disordered.

Furthermore, we acknowledge the emerging potential of AlphaFold3 and have incorporated a forward-looking perspective into our discussion. This includes our plans to explore and potentially integrate AlphaFold3 into our methodology, aiming to harness its advanced capabilities to enhance the accuracy and applicability of MissenseNet.

We believe that addressing these aspects comprehensively in the Conclusions section not only provides a clear summary of our work but also sets the stage for future research directions. We hope this clarification meets your expectations and further enhances the manuscript's impact.

Thank you once again for your insightful feedback, which is instrumental in refining our presentation and discussion of MissenseNet.

Reference:

[1]  Fang G, Jun H, Yiheng Z, Dongjun Y. Review on pathogenicity prediction studies of non-synonymous single nucleotide variations. Journal of Nanjing University of Science And Technology. 2021;45:1-17. doi:10.14177/j.cnki.32-1397n.2021.45.01.001

[2]  Adzhubei IA, Schmidt S, Peshkin L, Ramensky VE, Gerasimova A, Bork P, et al. A method and server for predicting damaging missense mutations. Nat Methods. 2010;7(4):248-249. doi:10.1038/nmeth0410-248

[3]  Abramson J, Adler J, Dunger J, Evans R, Green T, Pritzel A, et al. Accurate structure prediction of biomolecular interactions with AlphaFold 3. Nature. 2024;630(8016):493-500. doi:10.1038/s41586-024-07487-w

[4]  Karczewski K, Francioli L, Tiao G, Cummings BB, Alföldi J, Wang QS, et al. The mutational constraint spectrum quantified from variation in 141,456 humans. Nature. 2020;581(7809):434-443. doi:10.1038/s41586-020-2308-7

[5]  Landrum MJ, Lee JM, Riley GR, Jang W, Rubinstein WS, Church DM, et al. ClinVar: public archive of relationships among sequence variation and human phenotype. Nucl Acids Res. 2013;42(D1):D980-D985. doi:10.1093/nar/gkt1113

[6]  Schwarz JM, Cooper D, Schuelke M, Seelow D. MutationTaster2: mutation prediction for the deep-sequencing age. Nat Methods. 2014;11(4):361-362. doi:10.1038/nmeth.2890

[7]  Choi Y, Sims G, Murphy S, Miller J, Chan A. Predicting the Functional Effect of Amino Acid Substitutions and Indels. de Brevern AG, ed. PLoS ONE. 2012;7(10):e46688. doi:10.1371/journal.pone.0046688

[8]  Shihab HA, Gough J, Cooper DN, Stenson PD, Barker GLA, Edwards KJ, et al. Predicting the functional, molecular, and phenotypic consequences of amino acid substitutions using hidden Markov models. Human Mutation. 2013;34(1):57-65. doi:10.1002/humu.22225

[9]  Lu Q, Hu Y, Sun J, Cheng Y, Cheung K, Zhao H. A Statistical Framework to Predict Functional Non-Coding Regions in the Human Genome Through Integrated Analysis of Annotation Data. Sci Rep. 2015;5(1). doi:10.1038/srep10576

Reviewer 2 Report

Comments and Suggestions for Authors

Feature Extractions:
I struggle to understand what features the authors extract and how they do this. The first subsection (1.1) seems a rather descriptive text, and the table is just a tool list.

Dataset Composition:
Streamline the description of the dataset-building process. The current narrative is intricate due to frequent shifts between dataset type (training/test/validation), database name ( GnomAD, ClinVar) and mutation effects (likely benign, benign, deleterious).

Table 1:
The omission of EVE (10.1038/s41586-021-04043-8) among the sequence-based prediction methods is critical. Additionally, the table formatting is disrupted by a page break, making it difficult to understand.

Figure 1:
It would be helpful to allow the reader to individuate ShuffleNet and Squeezxe-and_excitation in the pipeline. It also appears that AlphaFold2 contributes to the datasets, yet no reference to this is made in section 2.1. Besides, increase the colour contrast in the figure, as white text on a bright background is hard to read.

Lines 34:
Given the audience, expanding on the phrase "potentially leading to various diseases" is almost mandatory. Add details in relevant fields such as nutrition (I suggest refs 10.1007/BF03256449, 10.1007/S12263-010-0186-6), cancer (I suggest refs 10.1158/0008-5472.c.6495380, 1422-0067/22/11/5416), and rare diseases (I suggest refs 10.1186/s12859-018-2416-7, 10.1093/hmg/ddv309).

Lines 103/112:
Use "consensus" or "ensemble" rather than "comprehensive."

Lines 140-142:
Clarify if the authors run predictions for mutants when they mention "predicted using AlphaFold2." Be aware that "AlphaFold is largely insensitive to input sequence variation and cannot accurately predict structural changes upon point mutation" (AlphaMissense paper). Additionally, given that this is its first appearance, explain the term "FEATURE" if it refers to the authors' framework or provides relevant references.

Lines 167-169:
I spot a potential issue. "One star" does not discriminate between single submitter and a conflicting classification. Are the authors filtering out the latter?

Lines 272-276:
This section is misplaced in the methods section. Move it to the conclusion and tone down the language.

Author Response

Comments and Suggestions for Authors
2:

Question 1:Feature Extractions:
I struggle to understand what features the authors extract and how they do this. The first subsection (1.1) seems a rather descriptive text, and the table is just a tool list.

Response:

Dear Reviewer,

Thank you for your feedback on our manuscript, particularly concerning the clarity of the feature extraction methods described in subsection 1.1. We apologize for any confusion caused by the placement of the detailed description of our feature extraction techniques, which was inadvertently located in section 2.2 under Data Acquisition.

We have taken your comments into consideration and have now revised subsection 1.1 to explicitly detail the feature extraction methods used in our study. Additionally, we have updated Table 1 to clearly delineate the predictive features ultimately employed in our analyses, specifying the role and utility of each tool listed.

We appreciate your guidance in enhancing the clarity and comprehensiveness of our manuscript and believe that these revisions will make the methodology more accessible and understandable for our readers.

Thank you again for your constructive comments. We look forward to your further suggestions.

Question 2:Dataset Composition:
Streamline the description of the dataset-building process. The current narrative is intricate due to frequent shifts between dataset type (training/test/validation), database name (GnomAD, ClinVar) and mutation effects (likely benign, benign, deleterious).

Response:

Dear Reviewer,

Thank you for your feedback on the original complexity of the dataset description. Following your recommendations, we have revised the section to provide a more streamlined and coherent explanation of our dataset construction.

The revised section now reads:

"Three datasets of missense variants have been compiled from the gnomAD (version 3.1) and ClinVar (version 20210131) databases to support the training, validation, and testing phases of the study. Clinically annotated non-synonymous variants are categorized as 'Benign' or 'Pathogenic' to represent benign and pathogenic variants, respectively. Eighty percent of these non-synonymous variants were randomly selected from proteins, with gnomAD variants designated for the training set and ClinVar variants not present in gnomAD allocated to the validation set. The remaining 20% of proteins containing ClinVar variants have been designated for the test set. To enhance the accuracy and reliability of the research data, all variants in the ClinVar dataset with a review status of less than one star have been excluded."

We believe this revision clarifies the dataset compilation process and enhances the manuscript's readability and accessibility. We appreciate the opportunity to improve our work based on your invaluable suggestions.

Question 3:Table 1:
The omission of EVE (10.1038/s41586-021-04043-8) among the sequence-based prediction methods is critical. Additionally, the table formatting is disrupted by a page break, making it difficult to understand.

Response:

Dear Reviewer,

Thank you for your meticulous review of our manuscript and for highlighting the issues concerning Table 1, specifically regarding the omission of the EVE tool1 and the table formatting.

Regarding the omission of EVE: The intent of Table 1 was to enumerate the sequence-based predictive features that were directly utilized in our research. While EVE is indeed a significant predictive method, it was not employed in our study to generate predictive features. However, recognizing the importance of EVE, we will include a reference to this tool in the relevant literature section to provide readers with a comprehensive understanding of its capabilities and applications. In the revised manuscript, we will add the reference (10.1038/s41586-021-04043-8) and discuss the context of EVE’s application in sequence-based prediction methods.

Regarding the table formatting issue: We apologize for the difficulties caused by the table formatting due to page breaks—an oversight during the document layout phase. We have since restructured the table to ensure that it is presented cohesively on a single page in the revised manuscript, thereby improving readability and preventing any disruption in understanding the data.

We greatly appreciate your feedback, which has been instrumental in enhancing the quality and clarity of our paper. We have made the necessary revisions and eagerly await your further comments on the updated manuscript.

Question 4:Figure 1:
It would be helpful to allow the reader to individuate ShuffleNet and Squeezxe-and_excitation in the pipeline. It also appears that AlphaFold2 contributes to the datasets, yet no reference to this is made in section 2.1. Besides, increase the colour contrast in the figure, as white text on a bright background is hard to read.

Response:

Dear Reviewer,

Thank you for your valuable feedback on Figure 1 in our manuscript. We appreciate your input and have made the following revisions to address your concerns:

We apologize for the initial lack of clarity in distinguishing between ShuffleNet 2and Squeeze-and-Excitation 3 within our pipeline. We have now revised Figure 1 to include detailed descriptions of both ShuffleNet and Squeeze-and-Excitation, ensuring that each component is clearly identified and visually distinct.

To clarify, the details concerning the methods associated with AlphaFold2 structural features are presented in Subsection 1.2, which is appropriately titled "AlphaFold2 Structural Features." Additionally, to maintain focus and clarity, Section 2.1 is dedicated exclusively to the composition of the datasets.

In response to your comments on the visual clarity of Figure 1, we have revised the figure to enhance the contrast between the text and the background, making it easier to read.

We believe these changes adequately address your concerns and enhance the manuscript's presentation. Thank you again for your constructive critique, which has significantly helped in refining our work.

Question 5:Lines 34:
Given the audience, expanding on the phrase "potentially leading to various diseases" is almost mandatory. Add details in relevant fields such as nutrition (I suggest refs 10.1007/BF03256449, 10.1007/S12263-010-0186-6), cancer (I suggest refs 10.1158/0008-5472.c.6495380, 1422-0067/22/11/5416), and rare diseases (I suggest refs 10.1186/s12859-018-2416-7, 10.1093/hmg/ddv309).
Response:

Dear Reviewer,

Thank you for your review and valuable feedback on our manuscript. Following your suggestions, we have expanded the section discussing how "various diseases" can be potentially caused by genetic variants, including detailed types of diseases and supported by relevant scientific references.

The revised text is as follows:

"Genetic variations, including non-synonymous single nucleotide variants (nsSNVs), have been extensively studied and found to be closely associated with multiple diseases. In the field of nutrition, poor nutritional status can exacerbate inflammatory responses and disrupt metabolic pathways, increasing the risk of cardiovascular diseases and diabetes4,5. In cancer research, certain genetic variations have been identified as increasing the risk of developing breast cancer and colorectal cancer 6,7. Additionally, genetic variations are linked to the occurrence of rare diseases such as Huntington's disease and cystic fibrosis8,9."

We hope these additions adequately respond to your suggestions and enhance readers' understanding of the relationship between genetic variations and diseases. We will continue to refine our manuscript to meet publication standards.

We look forward to your further guidance and feedback on the revised draft.

Question 6:Use "consensus" or "ensemble" rather than "comprehensive."

Response:

Dear Reviewer,

Thank you for your suggestion to use "consensus" or "ensemble" instead of "comprehensive" in line 103/112 of our manuscript. We appreciate your attention to the precise terminology that should be used to better convey the methodologies and results of our research.

Upon reflection, we agree that "consensus" or "ensemble" more accurately describes the methodology employed, where multiple models or data sources are integrated to form a unified conclusion. Using "comprehensive" might imply a broader scope than is intended in the context of our study.

We will revise the manuscript accordingly to reflect this more accurate terminology, ensuring that the description aligns better with the specific processes and outcomes involved.

Thank you once again for your guidance. We look forward to further improving our manuscript with your insightful feedback.

Question 7:Lines 140-142:
Clarify if the authors run predictions for mutants when they mention "predicted using AlphaFold2." Be aware that "AlphaFold is largely insensitive to input sequence variation and cannot accurately predict structural changes upon point mutation" (AlphaMissense paper). Additionally, given that this is its first appearance, explain the term "FEATURE" if it refers to the authors' framework or provides relevant references.
Response:

Dear Reviewer,

Thank you for your meticulous review and the critical observations you've raised regarding our use of AlphaFold2 for predicting structural changes in mutant proteins. Your insights are invaluable for improving the accuracy of our methodology description.

When we mentioned "using AlphaFold2 for predictions" in the text, it specifically referred to employing the AlphaFold2 model to predict the structure of the original (non-mutant) proteins, not direct predictions of structural changes post-mutation. As noted in the AlphaMissense10 paper you referenced, AlphaFold2 is not sensitive to minor sequence variations, such as point mutations, and therefore is not suited for directly predicting changes in such structures. We used the structural outputs from AlphaFold2 as a foundational basis to further analyze potential structural and functional impacts.

We sincerely apologize for the confusion caused by our description related to the term "FEATURE11." It was indeed an oversight, as "FEATURE" refers to an existing tool rather than a framework developed by us. This tool was employed to extract data from AlphaFold2's structural predictions for our analysis.

Regrettably, our initial manuscript did not properly cite or explain this usage, which may have led to misunderstanding. We will correct this error in the revised manuscript to ensure clarity and proper attribution.

We are extremely grateful for your detailed review and the essential issues you have pointed out. We look forward to addressing these points and refining our manuscript according to your valuable feedback.

Question 8:Lines 167-169:

I spot a potential issue. "One star" does not discriminate between single submitter and a conflicting classification. Are the authors filtering out the latter?

Response:

Dear Reviewer,

First and foremost, thank you for your meticulous review and the invaluable feedback you provided regarding our manuscript. Regarding your query about the "one star" classification in the ClinVar database and its differentiation between a single submitter and conflicting classifications, we acknowledge that this was an oversight on our part.

In the ClinVar database, the "one star" status (single submitter - criteria provided) indeed does not distinguish between conflicting classifications. This classification system merely reflects whether the submitter has provided evaluation criteria and does not directly address or annotate conflicts between classifications. This was an aspect that we did not sufficiently clarify in our paper, and we appreciate your pointing it out.

At the onset of our experimental study, the most recent version of the ClinVar dataset (version 20210131) was employed. The choice to use data rated at least "one star" primarily stems from the fact that such data provides evaluation standards from at least one submitter. Although this may not represent the most ideal high-consensus data, it offers a baseline reference for our research. Furthermore, as indicated by Ghosh R, Oak N, Plon SE in their 2017 study published in Genome Biology, titled "Evaluation of in silico algorithms for use with ACMG/AMP clinical variant interpretation guidelines,"12 many studies also opt for "one-star" data 13,14. This indicates that the selection is both commonplace and grounded in practical application, providing additional support and justification for our choice of such data.

Thank you again for your attention to our work and your suggestions. We will clarify this point in the revised manuscript and provide a more detailed discussion on the use and limitations of the ClinVar database in the discussion section.

Question 9:Lines 272-276:
This section is misplaced in the methods section. Move it to the conclusion and tone down the language.

Response:

Dear Reviewer,

Thank you for your insightful comments regarding the placement and tone of the text in lines 272-276. Our intention was indeed to emphasize the strategic enhancements made to ShuffleNet V2 and how these improvements support the development of Mis-senseNet, particularly its enhanced predictive accuracy for pathogenicity in missense variants.

We acknowledge your suggestion to relocate this section to the conclusion and moderate the tone. While we agree that the tone needs adjustment to avoid overstatement, we believe that the content is crucial in the section 3.2 on ShuffleNet V2, as it underscores the technical advancements and their relevance to the overall model development.

Therefore, we plan to retain a revised version of this content in section 3.2 to maintain the flow and integrity of the technical discussion. Additionally, we will revisit this topic in the conclusion with a more measured tone, reinforcing its significance in a broader context and ensuring it aligns with the overall narrative of our study.

We hope this approach addresses your concerns while preserving the essential elements of our manuscript. We look forward to your further guidance and feedback.

Reference:

[1]   Frazer J, Notin P, Dias M, Gomez AN, Min JK, Brock KP, et al. Disease variant prediction with deep generative models of evolutionary data. Nature. 2021;599(7883):91-95. doi:10.1038/s41586-021-04043-8

[2]   Zhang X, Zhou X, Lin M, Sun J. ShuffleNet: An Extremely Efficient Convolutional Neural Network for Mobile Devices. In: Proceedings of the IEEE Conference on Computer Vision and Pattern Recognition (CVPR). ; 2018.

[3]   Hu J, Shen L, Sun G. Squeeze-and-Excitation Networks. In: Proceedings of the IEEE Conference on Computer Vision and Pattern Recognition (CVPR). ; 2018.

[4]   Ferguson L. Nutrigenomics. Mol Diag Ther. 2006;10(2):101-108. doi:10.1007/BF03256449

[5]   Kaput J, Evelo C, Perozzi G, Ommen B, Cotton R. Connecting the Human Variome Project to nutrigenomics. Genes Nutr. 2010;5(4):275-283. doi:10.1007/s12263-010-0186-6

[6]   Kaminker J, Zhang Y, Waugh A, Haverty PM, Peters B, Sebisanovic D, et al. Distinguishing cancer-associated missense mutations from common polymorphisms. Cancer Res. 2007;67(2):465-473. doi:10.1158/0008-5472.CAN-06-1736

[7]   Petrosino M, Novak L, Pasquo A, Chiaraluce R, Turina P, Capriotti E, et al. Analysis and Interpretation of the Impact of Missense Variants in Cancer. IJMS. 2021;22(11):5416. doi:10.3390/ijms22115416

[8]   Cimmaruta C, Citro V, Andreotti G, Liguori L, Cubellis M, Mele BH. Challenging popular tools for the annotation of genetic variations with a real case, pathogenic mutations of lysosomal alpha-galactosidase. BMC Bioinformatics. 2018;19(S15). doi:10.1186/s12859-018-2416-7

[9]   Turner TN, Douville C, Kim D, Stenson PD, Cooper DN, Chakravarti A, et al. Proteins linked to autosomal dominant and autosomal recessive disorders harbor characteristic rare missense mutation distribution patterns. Hum Mol Genet. 2015;24(21):5995-6002. doi:10.1093/hmg/ddv309

[10] Cheng J, Novati G, Pan J, Bycroft C, Žemgulytė A, Applebaum T, et al. Accurate proteome-wide missense variant effect prediction with AlphaMissense. Science. 2023;381(6664). doi:10.1126/science.adg7492

[11] Halperin I, Glazer DS, Wu S, Altman R. The FEATURE framework for protein function annotation: modeling new functions, improving performance, and extending to novel applications. BMC Genomics. 2008;9(S2):S2-S2. doi:10.1186/1471-2164-9-S2-S2

[12] Ghosh R, Oak N, Plon S. Evaluation of in silico algorithms for use with ACMG/AMP clinical variant interpretation guidelines. Genome Biol. 2017;18(1). doi:10.1186/s13059-017-1353-5

[13] Wright C, Eberhardt RY, Constantinou P, Hurles M, FitzPatrick D, Firth H. Evaluating variants classified as pathogenic in ClinVar in the DDD Study. Genetics in Medicine. 2020;23(3):571-575. doi:10.1038/s41436-020-01021-9

[14] Zhang H, Xu MS, Chung WK, Shen Y. Predicting functional effect of missense variants using graph attention neural networks.

Round 2

Reviewer 1 Report

Comments and Suggestions for Authors

No comments